# Both trust in, and polarization of trust in, relevant sciences have increased through the COVID-19 pandemic

Sofia Radrizzani[1], Cristina Fonseca[2], Alison Woollard[3], Jonathan Pettitt[4], Laurence D. Hurst[1]*

1 The Milner Centre for Evolution, Department of Life Sciences, University of Bath, Bath, Somerset, United Kingdom, 2 The Genetics Society, London, United Kingdom, 3 Biochemistry Department, University of Oxford, Oxford, United Kingdom, 4 School of Medicine, Medical Sciences and Nutrition, University of Aberdeen, Institute of Medical Sciences, Aberdeen, United Kingdom

* l.d.hurst@bath.ac.uk

**Data Availability Statement:** All R scripts and data are available from https://github.com/ldhurst/Change_in_trust.git.

## Abstract

While attempts to promote acceptance of well-evidenced science have historically focused on increasing scientific knowledge, it is now thought that for acceptance of science, trust in, rather than simply knowledge of, science is foundational. Here we employ the COVID-19 pandemic as a natural experiment on trust modulation as it has enabled unprecedented exposure of science. We ask whether trust in science has on the average altered, whether trust has changed the same way for all and, if people have responded differently, what predicts these differences? We 1) categorize the nature of self-reported change in trust in "scientists" in a random sample of over 2000 UK adults after the introduction of the first COVID vaccines, 2) ask whether any reported change is likely to be real through consideration of both a negative control and through experiment, and 3) address what predicts change in trust considering sex, educational attainment, religiosity, political attitude, age and pre-pandemic reported trust. We find that many more (33%) report increased trust towards "scientists" than report decreased trust (7%), effects of this magnitude not being seen in negative controls. Only age and prior degree of trust predict change in trust, the older population increasing trust more. The prior degree of trust effect is such that those who say they did not trust science prior to the pandemic are more likely to report becoming less trusting, indicative of both trust polarization and a backfire effect. Since change in trust is predictive of willingness to have a COVID-19 vaccine, it is likely that these changes have public health consequences.

## Introduction

People differ greatly in their attitudes to well-evidenced science [1, 2], sometimes with consequences for both the individual and for the welfare of others (e.g. vaccine uptake [3]). With a view to promoting science, there is an extensive history of attempts to understand determinants of this variation in attitudes to science (see e.g. [1, 4, 5]). Early analyses focused on the relationship between science knowledge and attitude. Indeed, programs of research set in train

**Funding:** This work was funded by The Genetics Society.

**Competing interests:** The authors have declared that no competing interests exist.

in the 1980s repeatedly find that the level of "textbook" scientific knowledge is positively correlated with attitude (for review see [1]). This result underpins the deficit model of science communication [1, 2] and provided the rationale for Public Understanding of Science (PUS) activities that focused on attempts to increase scientific knowledge and literacy [1].

While some data are supportive of a possible causal link between knowledge gain and attitudinal change (see e.g. [6–11]) knowledge gain is unlikely to be the sole arbiter of attitude [12–17]. Indeed, in experimental cases where attitude has changed following instruction, it is not clear that instruction was causative. For example, although students that show larger increases in understanding of both genetics and evolution show larger increases in evolution acceptance [9], on qualitative follow-on analysis the accepting students could not provide evidence for evolution to support their case [9], but instead referred to endorsement from trusted figures, including teachers. This suggests that the mere fact of teaching can lead to increased acceptance owing to implicit endorsement, rather than knowledge gain *per se* [9].

The deficit model has more generally been criticised because it marginalizes potentially important externalities [18]. In the context of politicized science (vaccines, stem cells, climate change), for example, many assumptions of the deficit model break down [4, 5] (but see also [19]) with high scientific literacy being associated with both extremes of position [5]. In this alternative model, the facts of the science *per se* are seen as secondary, the religio/political/ethical position providing the scaffolding for the view on the aligned science [4, 5]. Engaging with such externalities is thus seen as important to move beyond the deficit model [20].

Failure to allow for externalities can lead to a back-firing effect in which correction exposure increases belief in the relevant misperception among individuals predisposed to believe the claim [21]. In the UK, a similar attitudinal backfire effect is thought to be a consequence of a series of science communication failures (the BSE crisis and the GM food debate) that led to an increase in negative attitudes to science [22]. Stimulated by notions of a public crisis of confidence in science, science communication strategies shifted from a focus on public deficits in knowledge to a Science-in-Society approach that highlighted a trust deficit [22, 23]. This led to advocacy for science events (hearings, citizen juries, science festivals) with the aim of building trust [22]. Since then, there has been much discussion of the importance of trust [24–27], what trust is (see e.g. [28]), what might mediate trust (see e.g. [3, 28, 29]) and what the correlates of trust might be (see e.g. [30]), with attempts to infer causality [31, 32].

In this context, the COVID-19 pandemic presents an unusual, albeit uncontrolled, opportunity [33]. Countering the epidemic required not only extensive between-country scientific cooperation [34] but communication of the science from both government and scientists [34]. In many regards, each country presents itself as an independent experiment on trust modulation.

Early studies during the COVID pandemic, prior to vaccine development, found no evidence for an increase in trust in either the UK [35] or US [36]. However, comparison of New Zealanders during and prior to lockdown report higher levels of trust in science in the former [37]. Here we return to these issues employing survey data from a demographically representative group (N>2000) of the UK population. The survey was conducted in June 2021 and hence several months after the introduction of the first vaccines.

Specifically, we start by asking whether this increased exposure to science is associated with altered trust in science. We find strong evidence for such an effect. We then ask whether this can in any manner be more directly related to relevant science activities through the pandemic. To this end we consider a negative control (trust in geologists) and provide an experiment in which half of respondents are asked about their change in trust in the pharmaceutical industry using a player in the vaccine industry (Pfizer) or one not (GlaxoSmithKline) as examples. We find robust evidence that increase in trust is specially associated with the relevant players.

Having established a net increase in trust, we then ask whether trust changed the same way for all, finding against such a hypothesis. Given this, we then ask what predicts these differences. Trust in scientists is, for example, predicted by age [35], religiosity [38], educational attainment [31] and gender [31, 35]. Do these same characteristics also then predict any tendency to change trust and does the level of trust prior to the pandemic predict the change in trust when controlling for these covariates? We find an increased variance in net trust with those more negative prior to the pandemic tending to become even more negative through it and those initially more positive becoming even more positive. This is one form of polarization [39]. Finally, we ask whether change in trust is predictive of relevant behaviours, in this case uptake of the COVID-19 vaccines (cf. [27, 40–42]). Increase in trust predicts a propensity to take the vaccine.

## Materials and methods

### The survey

The survey was funded by The Genetics Society and delivered through Kantar Public (Kantar Public's random sample panel Public Voice, panel survey 11, June 2021). The target group for the survey were UK individuals above the age of 18 who lived in residential accommodation. Respondents were paid. After Quality Control of responses, Kantar Public provided survey data on over 2000 people through both online and telephone means (to not exclude individuals without access to the internet).

The survey design maximises the geodemographic representativeness of any sample drawn from the panel as well as minimising the number of households in which more than one panel member is selected for the same survey (see S1 File). One measure of the representativeness of a respondent sample is its 'weight efficiency' after it has been calibrated to a benchmark. In this case, the benchmark is the weighted recruitment survey respondent sample, standing in for the true target population (i.e. all UK-resident individuals aged 18+ living in private accommodation). A perfectly representative sample will have a weight efficiency of 100%, indicating that no variance in response probabilities was observed. The weight efficiency for the dataset was 72%. This is substantially higher than the equivalent for the recruitment surveys taken together so there was some gain in representativeness between the recruitment surveys and this panel survey. This was largely due to the sampling protocol adopted (i.e. different sampling fractions applied to each selection stratum). Furthermore, sampling and non-response between the recruitment survey and the panel survey reduced the mean number of cases per sample cluster. This had a positive knock-on effect on the overall statistical efficiency of panel survey, as measured by the recruitment survey variables. On this basis, the relative overall statistical efficiency of the survey dataset (compared to the recruitment survey dataset) was 177%. For more detail see S1 File. Given this, no further adjustment for bias was deemed necessary.

Questions for the survey were drawn (or minorly adapted) from prior benchmarked surveys and pre-approved by Kantar Public as fit for purpose. As a consequence, a pilot survey was deemed unnecessary. The data was not collected to specifically test this hypothesis and so was not pre-registered. Full design of questions, answers and sociodemographic information are included in the supplementary material (see S1 File). Here, the key questions required to conduct the statistical study are explained, along with the respective scoring system used for analysis purposes.

### Analysis strategy

**Has attitude changed over the COVID-19 pandemic?.** Firstly, respondents were asked in what way their trust towards scientists had changed with respect to the start of the pandemic. They were asked:

"Would you say you now trust scientists more, less, or about the same as you did at the start of the pandemic?"

The possible answers were: "Trust them much less", "Trust them a little less", "About the same", "Trust them a little more" and "Trust them much more". Each response was scored on a scale from -2 to +2, where negative values were representative of a decrease in trust, 0 to no change, and positive values to an increase in trust. Respondents had the option to not answer, in this case the results are scored as "NA" and not included in the statistical analysis.

An identical structure was applied to questions related to "geneticists" and "geologists". Geologists are treated as a negative control.

Participants were also questioned about their change in trust in pharmaceutical companies, with identical formats as questions regarding scientists, geneticists and geologists, respondents being asked:

Would you say you now trust pharmaceutical companies, e.g. XXX, more, less or about the same as you did at the start of the pandemic?

The answers were also scored from -2 (decrease in trust) to +2 (increase in trust). Within the question, an example of a pharmaceutical company was included (XXX). As an embedded experiment, half of the respondents were exposed to a question using "Pfizer" as an example, while the other half "GlaxoSmithKline" (GSK). These companies were selected as, at the time when the survey was conducted, Pfizer had gained recognition as a vaccine provider, whilst GSK had no presence in the media. The results between respondent groups were compared to give *prima facie* evidence on whether a corporation's effort of delivering a vaccine, and the media exposure associated with it, could have impacted trust.

**Has attitude polarized over the COVID-19 pandemic?.** Respondents were asked to think back to before the start of the pandemic and express what their opinion on a statement explicitly mentioning trust would have been. People were asked to what extent they would have agreed with the statement "those in charge of new developments in genetic science cannot be trusted to act in society's interests". Results were scored -2 (strongly agreed) to +2 (strongly disagreed) on a five-point Likert scale.

Respondents then expressed how their view towards the statement had changed over the pandemic being asked:

"How do you think your views have changed in the last year? Thinking about the statement "Those in charge of new developments in genetic science cannot be trusted to act in society's interests" would you say that you have. . ."

This was scored on a three-point scale: more likely to agree (+1), no change (0), more likely to disagree (-1). The change in trust scores (-1 to +1) were added to the respondents' respective pre-pandemic trust scores (-2 to +2) to find their post-pandemic trust (-3 to +3).

The variance of pre- and post-pandemic trust datasets was compared as a measure of polarization [39]. Specifically, the scores for each individual were added from which a net variance could be calculated. This was then compared with a variance expected under null in which one of the two vectors was randomized and added to the non-randomized vector (repeated a million times). The variance of each of these null distributions was calculated and distribution of the null variances compared with the observed variance to determine the effect of the pandemic on the variance. Three models were hypothesised to predict a change in variance. Firstly, model A assumes variance has remained unchanged (null hypothesis). Secondly, model B infers a decrease in variance, representative of a more uniform opinion on science among the public. Conversely, model C hypothesises an increase in variance.

**Is attitudinal change predictive of behaviour?.** To test whether change in attitude towards science had translated to behaviour, respondents were asked whether they would take a COVID-19 vaccine if they were offered one. A score of +1 was attributed to both people who

had already been vaccinated, and ones who expressed they would be willing to but still had not received it. This ensured that the results were not affected by the UK vaccination plan, as younger individuals were still not offered the possibility to get vaccinated at the time the survey was conducted. Also, this avoided confusion in people who had already received the vaccine, as they could have expressed that they would not be willing to receive an additional dose. People who expressed they would not accept the vaccine were scored with 0.

## Covariate data

Sociodemographic information was monitored alongside and used to control for covariates. All covariates were treated as independent variables. Firstly, respondent age was recorded directly as the numeric value. Respondents also indicated their sex, which, for analysis purposes, was scored with a binomial system (0 for males, 1 for females). Educational attainment was recorded with three options: degree level qualification (scored with 2), non-degree level (1) and no qualification (0). Respondents were also grouped based on their religiosity: religious and actively practising individuals (scored with 2), religious but not practising (1) and non-religious (0). As regards the vaccine we also controlled for prior COVID infection. History of COVID-19 infection was considered where individuals that had previously contracted it were scored 1. This included people that had either confirmed by a test, a professional worker or by their own suspicions. People who never had COVID-19 were scored 0, and those who preferred not to disclose this information were not included in the statistical analysis. For analysis of trust, dependent variables were degree of trust prior to the pandemic and change in trust. For behavioural attitude to the COVID vaccine the covariates and trust measures were the independent variables, the attitude the dependent variable.

To assay political attitude, subjects were asked ten questions (not presented in this order):

1. Rich people can get away with breaking the law

2. People's working conditions and wages need strong legal protection

3. Major public services and industries should be in state hands

4. People in Britain should be more tolerant of those who lead unconventional lives

5. The government should redistribute income from the better-off to those who are less well off

6. The monarchy should be abolished

7. People today don't have enough respect for traditional British values

8. Business in this country is over-regulated by the government

9. People who break the law should be given tougher sentences

10. There should be fewer immigrants in this country

Respondents were asked whether they "Strongly agree", "Tend to agree", "Neither agree nor disagree", "Tend to disagree", "Strongly disagree" or "Don't know". Failure to respond was, like "Don't know" classified as NA. Responses were scored -2 to +2 with a more positive response being more right wing in inclination. For questions 1 to 6 agreement was classified as negative and disagreement more right wing (hence positive score). For questions 7 to 10 we scored in the inverse manner. Each individual was ascribed one mean score across the 10 questions (with NAs eliminated from calculation of the mean). This in turn was divided in 2 to provide our "political" scale from -1 (strongly left wing) to +1 (strongly right wing). We also

recovered voting pattern from the 2019 UK general election. Analysis of the political scores as a function of party voted for supports the metric in so much as right wing parties (DUP, Conservative, Brexit) have positive scores, more left leaning parties (labour, SNP, SDLP) are more negative and centrist parties (liberal democrats) are more centrally scoring (See S1 Fig).

## Statistics

All statistics were conducted in R (version 4.1.2). All tests were performed using non-parametric methods (e.g. Kruskall-Wallis, Mann-Whitney U test, Spearman rank correlation) given the non-normal distribution of much of the data. Covariate control was performed via Spearman partial correlations. Correlation matrices were produced using the rcorr function within the Hmisc package, and partial correlation was found using the pcor package. In addition, the vioplot package was used to create violin plots. Full R scripts and data are available from https://github.com/ldhurst/Change_in_trust.git. No data was excluded.

## Ethics statement

The survey was commissioned by The Genetics Society to be performed by Kantar Public. Kantar Public adheres to the following standards and industry requirements: Market Research Society (MRS) and ESOMAR (the global voice of the data, research and insights community) professional codes of conduct, ISO 20252 international market research quality standard, ISO 9001 international standard for quality management systems and the Data Protection Act 2018. Given that respondents had granted consent to Kantar Public to be enrolled on their panel, further ethics approval was not required by Kantar Public for this particular research, but the MRS code of conduct was followed which provides ethical guidelines for the industry. Participants were paid by Kantar Public and provided electronic consent. Respondents had to be over 18. Data were analysed anonymously. The University of Bath ethics committee determined that with Kantar Public's prior ethical approval, prior consent and data handling standards, any further ethical approval process was not required.

## Results

### Attitudinal shifts over the COVID-19 pandemic

**Trust in scientists has increased in 33% of people.**    A total of 2035 participants expressed in what way their trust towards scientists had changed with respect to the start of the pandemic. The majority (N = 1215, 60%) expressed no change in trust (Fig 1). However, 33% (673 respondents) reported an increase in trust, whilst 7% (147 people) reported a decrease.

There was a significant split between the number of people expressing an increase (N = 673) and those expressing a decrease in trust (N = 147) such that out of those reporting a change in attitude (N = 820), 82% expressed an increase in trust (binomial test, $P < 2.2 \times 10^{-16}$). Our data thus indicates that, through the pandemic, there has a been a net positive change in trust in scientists.

### The observed change in trust is robust to negative control

While the above provides good evidence that trust in "scientists" has changed (at least subjectively) through the pandemic, can we be confident that this is owing to actions during the pandemic? We take two approaches to address this. First, we consider a negative control. Second, we ask experimentally about the pharmaceutical industry with examples that either were or were not involved in vaccine production.

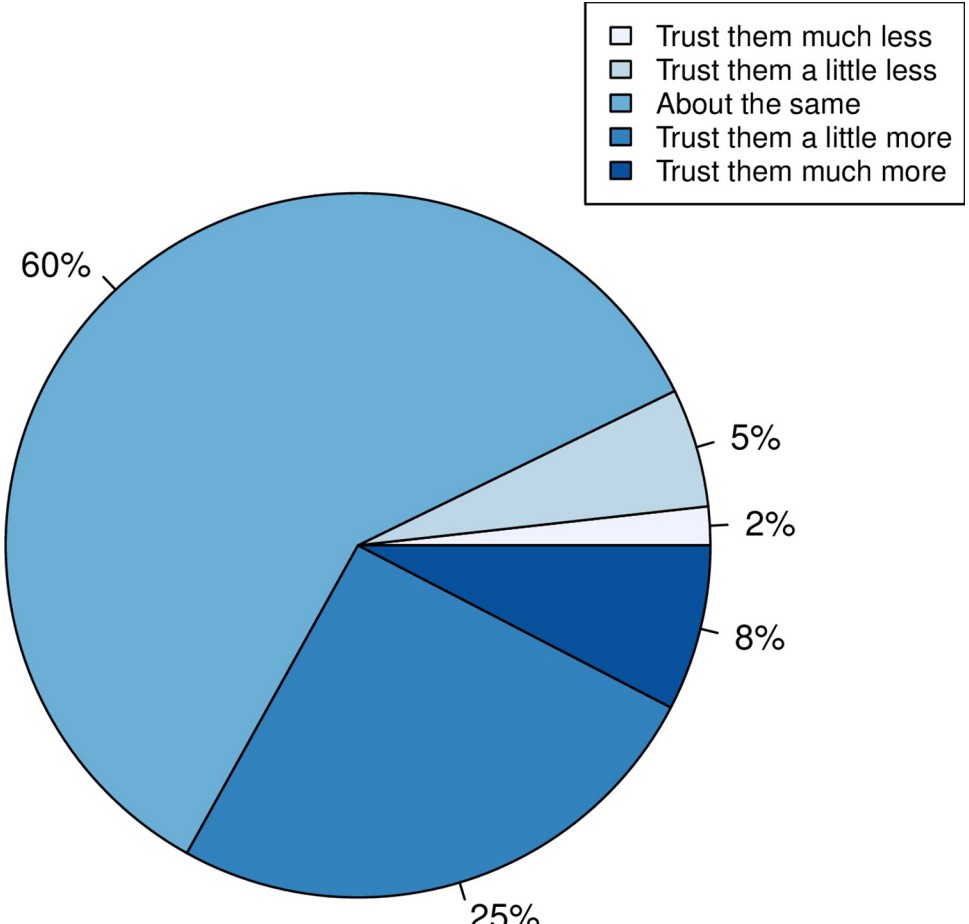

**Fig 1. Proportion of individuals in each class of self-reported change in trust in "scientists" during the COVID-19 pandemic (N = 2035).**

For a negative control we ask about trust in geologists. In addition, we ask about geneticists who were involved in the science but possibly not with as much recognition as "scientists". A general overview on the distribution of change in attitude towards geneticists and geologists was performed. For both groups, the majority of respondents showed no change in trust (79% for geneticists and 90% for geologists: Fig 2).

A three-way Kruskal-Wallis test reports a statistically significant difference between change in trust towards scientists, geneticists and geologists ($H(2) = 240.9$, $P<2.2 \times 10^{-16}$). A post-hoc Dunn test showed significant differences in each pairwise comparison both with no adjustment, and under Bonferroni correction ($P<2.2 \times 10^{-16}$ in both cases). Based on the assumption that attitude towards geologists acts as a negative control, this suggests that the pandemic had significant effects on trust towards geneticists, but most significantly "scientists" and that these effects cannot be dismissed as a general response to any "ologist".

## Trust in pharmaceutical companies has increased, in respect of their efforts/exposure over the pandemic

We asked whether people had altered their trust in pharmaceutical companies and provided them with one of two exemplars as part of an embedded experiment. A total of 1046 and 949

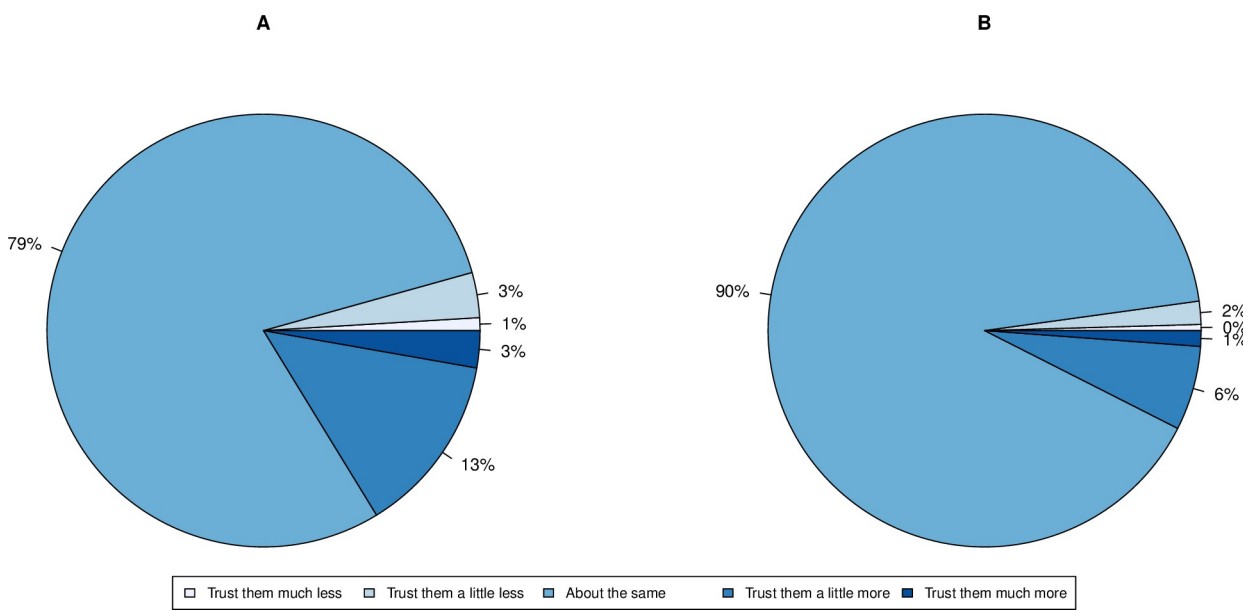

**Fig 2.** Proportion of individuals in each class of self-reported change in trust in A. "geneticists" (N = 1987) and B. "geologists" (N = 1969) during the COVID-19 pandemic. Figures are rounded such that < 0.5% is rounded to 0%.

respondents answered questions involving Pfizer and GSK respectively. In both cases, the majority (around 60%) expressed their trust had not changed. Out of those who expressed a change in trust (Pfizer N = 474, GSK N = 341), there were more people whose trust had increased (Pfizer N = 349, GSK N = 212), rather than decreased (Pfizer N = 125, GSK N = 129) (Fig 3). Binomial tests confirmed the difference was significant in both cases (Pfizer: P<2.2 x $10^{-16}$, GSK: P = 8.146 x $10^{-6}$).

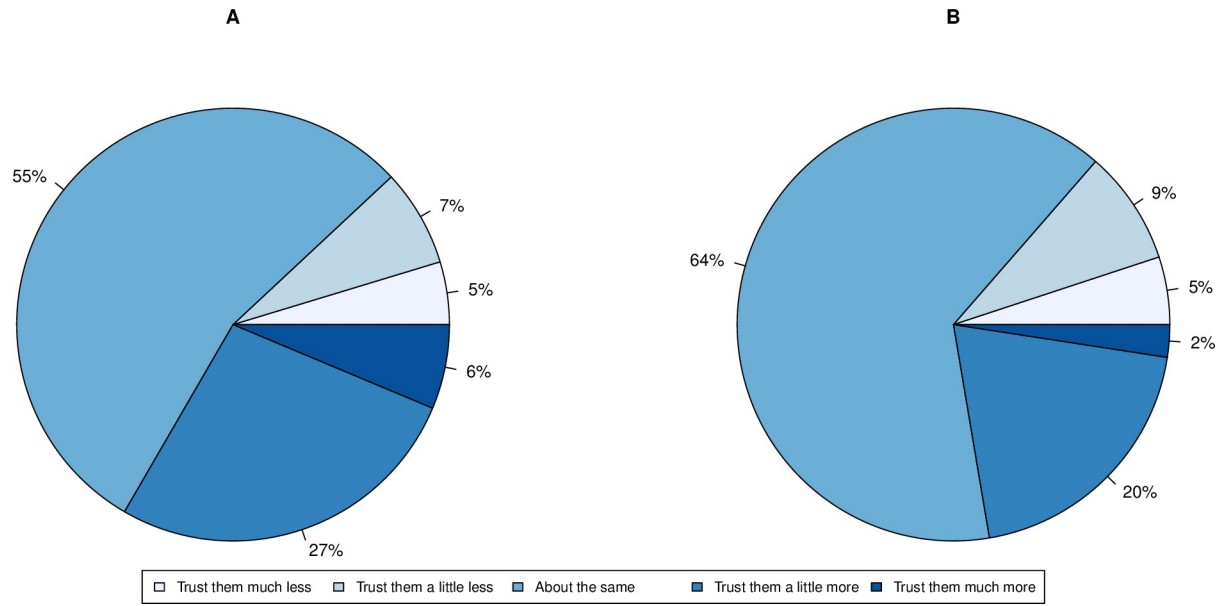

**Fig 3.** Proportion of individuals in each class of self-reported change in trust in pharmaceutical companies during the COVID-19 pandemic if given either A. Pfizer (N = 1046) or B. GlaxoSmithKline (N = 949) as exemplars.

A Chi-Squared goodness-of-fit test determined that the observed distribution of trust in Pfizer and GSK is not homogeneous ($X^2(4) = 36.13$, P = 2.718 x $10^{-7}$). Moreover, a Mann-Whitney U test confirmed a significant difference in change in trust between respondents who answered a question with Pfizer as an example, compared to when GSK was used (mean trust score in Pfizer group = 0.2304, mean trust score in GSK group = 0.0611, Mann-Whitney U test: P = 6.487 x $10^{-7}$, results were identical both with and without continuity correction). There is therefore sufficient evidence to conclude that the change in trust towards pharmaceutical companies was dependent on what corporation was mentioned, where naming "Pfizer" was associated with a more positive change in trust in pharmaceutical companies.

Individual Chi-Squared goodness-of-fit tests for each category of change in trust were informative in determining what the discrepancy between the two groups can be attributed to. Results show that the most significant difference lies within the categories of increase in trust ("Trust them a little more": $X^2(1) = 11.01$, P = 9.08x$10^{-4}$; "Trust them much more": $X^2(1) = 16.28$, P = 5.48x$10^{-5}$), as opposed to decrease in trust ("Trust them a little less": $X^2(1) = 0.92$, P = 0.338; "Trust them much less": $X^2(1) = 0.17$, P = 0.68). This confirms that there is a significantly larger number of people who expressed a positive change in trust within the Pfizer group, compared to the GSK group.

## Attitude has polarized with reference to pre-pandemic trust

**Evidence of polarization.** To ask about predictors of change in trust, we first tested for trust polarization, meaning an increase in variance in trust. We assayed self-reported change in trust and pre-pandemic trust levels. For each individual these two scores were added and a variance of the resulting vector calculated. Significance of this was evaluated via a randomization test (see Methods). We find that the observed variance (1.397) is larger than any in all of the one million simulations (hence P<1 x $10^{-6}$).

This increase in variance could be consistent with those originally positive becoming more positive while those originally negative have become more negative. Consistent with this, a Spearman's Rank Sum test shows a significant positive correlation between pre-pandemic trust and change in trust (Fig 4, $r_s$ = 0.31, P = 0.000, N = 2039). The correlation remains robust after controlling for the covariates age, sex, educational attainment, political attitude and religiosity (Table 1, partial $r_s$ = 0.31, P<0.0001, N = 2039). The change in trust is otherwise only predicted by age, with higher age weakly associated with a positive change in trust ($r_s$ = 0.06, P = 0.01, N = 2039). It is therefore possible to conclude that people reporting a more negative change in trust are mainly comprised of those who did not trust science prior to the pandemic either. Absolute trust levels currently reported are higher for those with higher education attainment ($r_s$ = 0.11, P< = 0.05), lower religiosity ($r_s$ = -0.05, P<0.05) and more left wing politics ($r_s$ = -0.06, P<0.05) but not by age ($r_s$ = -0.03, P>0.05). The correlation with political position is rendered non-significant on partial correlation.

The findings from the randomization test and the correlation analysis allow us to accept model C which suggests that people who had a negative attitude towards science prior to the pandemic decreased their trust and have become more negative. Similarly, people who already trusted science before the pandemic, now expressed a more positive attitude towards it. Although there has been an overall increase in trust, there is thus also evidence of polarization of trust.

## Attitudinal change is predictive of behaviour

**Decrease in trust predicts refusal to be vaccinated.** People were also questioned about their willingness to be vaccinated. Overall, 1930 respondents expressed they would accept a

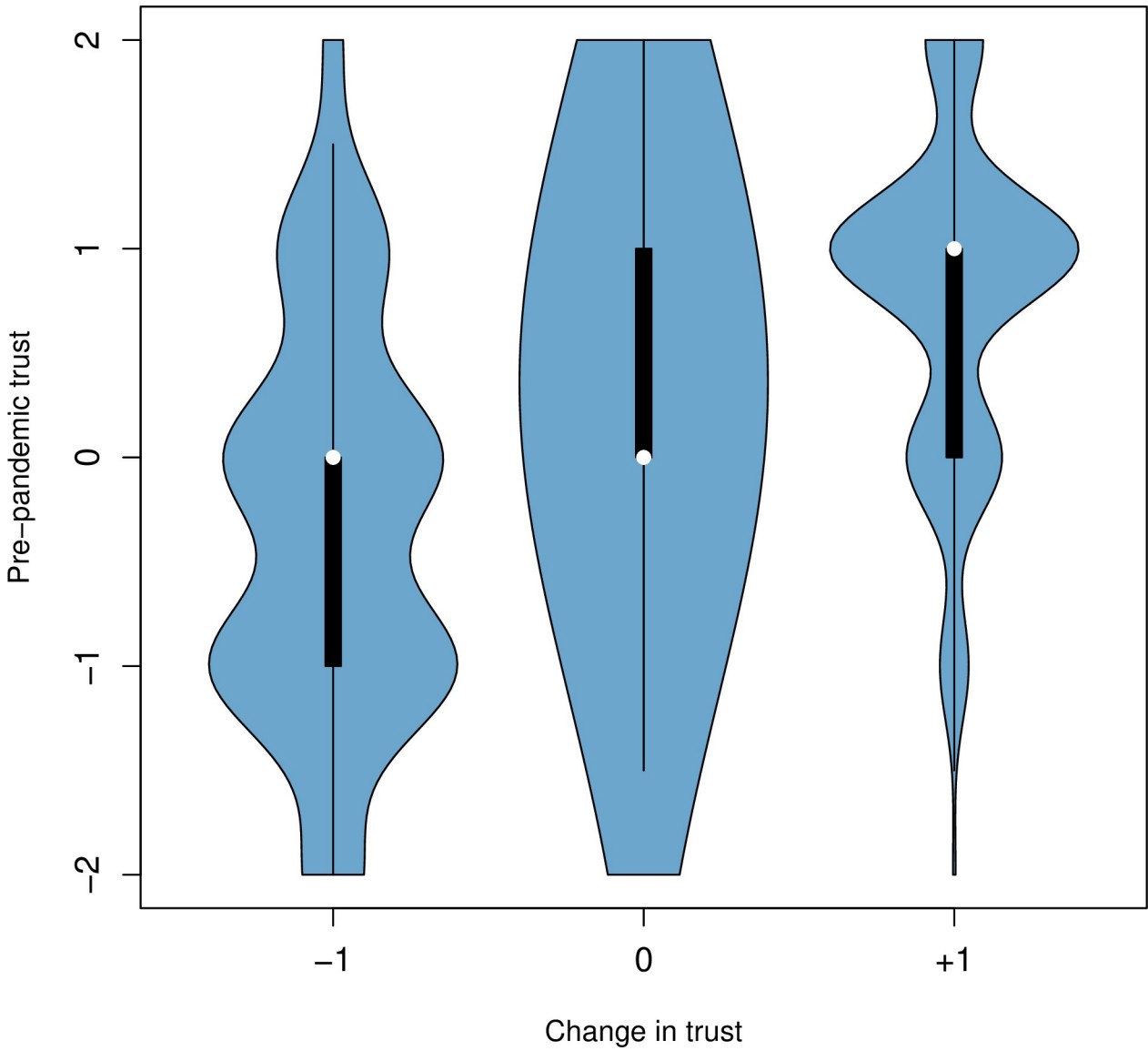

**Fig 4. Violin plot of the relationship between change in trust in scientists and self-reported pre-pandemic trust levels (N = 2039).**

COVID-19 vaccine if they were offered one, whilst 85 would not. Fig 5 shows the frequency of people who are willing or not willing to be vaccinated, in relation to their change in trust.

A Chi-Squared goodness-of-fit test finds that there is evidence of heterogeneity in change in trust ($X^2(1) = 53.37$, $P = 2.582 \times 10^{-12}$) between the two groups (will have vaccine versus will not). Furthermore, the results from the two vaccine groups are significantly different (mean trust score of those willing to accept a vaccine = 0.0601, mean trust score in those refusing to vaccine = -0.2706, Mann-Whitney U test: $P = 2.70 \times 10^{-8}$, results were identical with and without continuity correction). The difference seems to be driven by the people who reported their trust had gone down, as only the observed willingness to vaccine of this group is significantly different than expected ($X^2(1) = 46.06$, $P = 1.147 \times 10^{-11}$). This indicates that people who are not willing to get vaccinated have significantly less trust in science than those who would accept a COVID-19 vaccine.

**Table 1. Correlation and partial correlation between pre-pandemic trust score and change in trust score, controlling for age, sex, educational attainment, political attitude and religiosity.** Values above the diagonal are Spearman's rho values. Values below the diagonal are pairwise partial correlations controlling for the rest of variables. Values in bold are significant at P<0.05, N = 2039. Scoring system as follows: respondent age was recorded directly as the numerical value; sex was scored with a binomial system (0 = males, 1 = females); educational attainment recorded as degree level (+2), non-degree level (+1) or no qualification (0); religiosity recorded as: religious and practising (+2), religious but not practising (+1), non-religious (0). Politics is scored from -1 to +1, more positive implying more right wing based on 10 attitudinal questions (Methods).

| | Pre-pandemic trust | Change in trust | Age | Sex | Education | Religiosity | Politics |
|---|---|---|---|---|---|---|---|
| Pre-pandemic trust | - | **0.31** | -0.03 | -0.03 | **0.11** | **-0.05** | **-0.06** |
| Change in trust | **0.31** | - | **0.06** | -0.01 | 0.03 | 0.00 | -0.01 |
| Age | -0.24 | **0.07** | - | -0.02 | **-0.14** | **0.20** | **0.24** |
| Sex | -0.02 | 0.00 | -0.04 | - | -0.03 | **0.08** | -0.01 |
| Education | **0.10** | 0.00 | **-0.10** | -0.05 | - | 0.04 | **-0.26** |
| Religiosity | **-0.04** | 0.01 | **0.18** | **0.10** | **0.11** | - | **0.17** |
| Politics | -0.02 | -0.01 | **0.18** | -0.03 | **-0.24** | **0.15** | - |

To evaluate if this result is explained by covariates, a correlation and partial correlation approach was used. Spearman's Rank Sum test confirms a positive correlation between willingness to accept a vaccine and change in trust ($r_s$ = 0.12, P = 0.000, N = 1987). The correlation is robust of covariates age, sex, educational attainment, and religiosity, as well as previous history of contracting COVID-19 (Table 2, partial $r_s$ = 0.11, P = $6.88 \times 10^{-7}$, N = 1987). Of these other variables, willingness to receive a vaccine is otherwise predicted by age, where oldest individuals are more likely to get vaccinated (partial $r_s$ = 0.13, P = $5.16 \times 10^{-9}$, N = 1987). Sex could also be a predictor, where females are more likely to be vaccinated, although this correlation is not as significant (partial $r_s$ = 0.04, P = 0.05, N = 1987). The results allow us to conclude that refusal to be vaccinated is predicted by a decrease in trust.

## Discussion

In contrast to studies performed early in the COVID-19 pandemic in the UK [35] and the US [36], we find strong evidence for large net increase in trust in scientists over the COVID-19

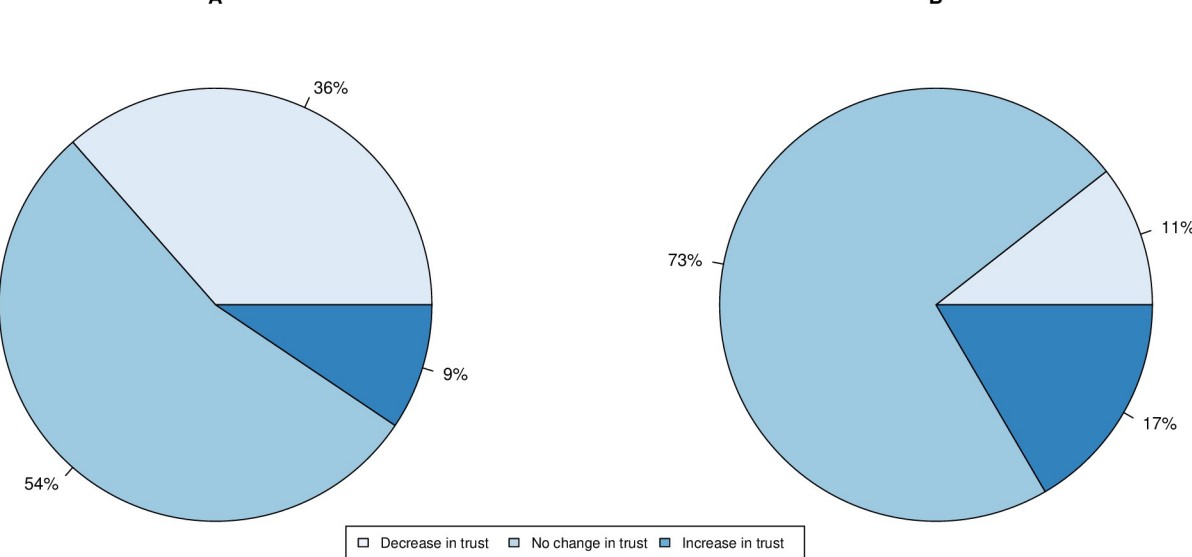

**Fig 5.** Proportion of individuals in each class of self-reported change in trust in scientists for A. those unwilling to be vaccinated (N = 85) and B. those willing to be vaccinated (N = 1930).

**Table 2. Correlation and partial correlation between willingness to take vaccine and change in trust, controlling for having previously contracted COVID-19, age, sex, educational attainment, religiosity and political inclination.** Reported figures in the upper panel are raw Spearman Rank correlations. In the lower panel they are partial Spearman's rho values, controlling for the other variables. Values in bold are significant at P<0.05, N = 1987. Scoring system as follows: COVID-19 recorded as: never have been infected (0), history of infection, either confirmed by a test, a professional worker or by their own suspicions (1); respondent age was recorded directly as the numerical value; sex was scored with a binomial system (0 = males, 1 = females); educational attainment recorded as degree level (+2), non-degree level (+1) or no qualification (0); religiosity recorded as: religious and practising (+2), religious but not practising (+1), non-religious (0). Politics is scored from -1 to +1, more positive implying more right wing based on 10 attitudinal questions (Methods).

|  | Vaccine | Change in trust | COVID-19 | Age | Sex | Education | Religiosity | Politics |
|---|---|---|---|---|---|---|---|---|
| Vaccine | - | **0.12** | **-0.07** | **0.14** | 0.04 | 0.03 | 0.02 | -0.01 |
| Change in trust | **0.11** | - | **-0.05** | **0.06** | 0.00 | 0.03 | 0.01 | -0.02 |
| COVID-19 | -0.03 | -0.03 | - | **-0.20** | -0.04 | -0.02 | 0.01 | -0.01 |
| Age | **0.13** | 0.04 | **-0.2** | - | -0.03 | **-0.15** | **0.20** | **0.24** |
| Sex | **0.04** | -0.01 | **-0.05** | **-0.06** | - | -0.04 | **0.09** | 0 |
| Education | 0.03 | 0.03 | **-0.05** | **-0.12** | **-0.06** | - | 0.04 | **-0.26** |
| Religiosity | -0.01 | 0.00 | **0.05** | **0.18** | **0.10** | **0.11** | - | **0.18** |
| Politics | -0.03 | -0.02 | 0.01 | **0.18** | -0.02 | **-0.24** | **0.155** | - |

pandemic. However, at the same time, people's trust has polarized compared to the beginning of the pandemic such that negative pre-pandemic attitudes were predictive of a decrease in trust. As trust in science is a key predictor of willingness to engage with public health measures, such as vaccination [3, 42], it is highly likely that the change in trust that we find has affected vaccine uptake. That the change in trust is predictive of willingness to accept a COVID-19 vaccine is consistent with this.

While this evidence suggests that something about the pandemic has had a large effect on the British public's trust in science, opposite to that seen through BSE and GM debates, the evidence does not address why public trust has gone up. Nonetheless, our further evidence is consistent with the notion that actions by scientists and science-companies are influential in the change in trust. We do not see the same change in trust in geologists as we do for "geneticists" or for "scientists". Testing trust towards pharmaceutical companies within the survey showed that a larger proportion of respondents reported an increase in trust upon the mentioning of "Pfizer", compared to "GlaxoSmithKline". At the time the survey was conducted, Pfizer had just released its mRNA vaccine, whilst GSK had not contributed to the management of the pandemic. At the time there were no suggestions of adverse reactions to the Pfizer vaccine (which is why Pfizer rather than AstraZeneca was chosen as the example). Whether the more positive attitude when Pfizer was mentioned was because of its development of a vaccine or its media exposure we cannot say. Note too that the question was worded so that respondents were asked about their change in trust in pharmaceutical companies with Pfizer and GSK mentioned as examples. Therefore, the results do not indicate a more positive change in trust in Pfizer but simply the fact that mentioning Pfizer in the question had triggered overall a more positive response.

Analysis of predictors of change in trust reports only two significant predictors, pre-pandemic trust and age (Table 1). It is notable that religiosity, educational attainment and sex do not predict change in trust. This suggests that level of trust and change in trust are demographically different. We are hesitant to extrapolate too much from the age effect as the COVID-19 epidemic was odd, especially affecting the old. Indeed, age is a strong predictor of whether individuals will take the vaccine in this same data [43] probably because they are also more at risk. Thus, we are left with one possibly generalizable result, namely that change in trust is well predicted by self-reported pre-pandemic degree of trust such that those that were more negative before the pandemic have become even more convinced of their opinion. Qualitative

follow up analysis of such individuals could prove especially valuable allowing one to ask the question, what, if anything would convince you to place more trust in science?

## Caveats and limitations

This polarization result, however, comes with an important caveat. We rely not on measures of trust prior to the pandemic versus during the pandemic [37]) but rather on self-assessed change in trust during the pandemic [35]. While this is not problematic in so far as we can be clear that our analysis concerns self-assessed change in trust, as opposed to more objectively measured change, it could be problematic for the analysis of polarization. People may, for example have inserted their current disposition towards science as their prior disposition. That is to say, those reporting that they have become less trusting may report that they always were less trusting by confusing their current emotional/attitudinal state and their recalled emotional/attitudinal state. Such an effect would predict the increased variance that we observe. Replication using data obtained prior to the pandemic would then be valuable even if this could not be done using matching paired samples as we do.

We also examine exclusively trust in scientists. This presents a limitation on our conclusions. Prior work has for example highlighted the role of trust in governmentally derived information and the social environment (family and friends) as being important through COVID-19 as regards vaccine uptake (e.g. in India [27]).

A further limitation comes from our sample being UK based. Some of our results might be more general. For example, in India trust in the healthcare system is reported to predict lesser degrees of vaccine hesitancy [44], which may be comparable to trust in science. However, some results, appear to be nation specific. For example, unlike some prior surveys (e.g. in India [44]), we find no evidence for religiosity as a predictor of vaccine uptake/hesitancy. In Germany, older age, higher educational attainment and being a man predicts increased willingness to be vaccinated against COVID [45]. We replicate this studies' age effect on vaccine uptake (Table 2), but find no significant effect of educational attainment (although the effect is on the edge of significance and in the same direction as reported in Germany, i.e higher educational attainment, higher vaccine uptake). We do detect a sex effect (just under the edge of significance), but ours is in the opposite direction to that in Germany [45]. We are thus hesitant to make general (i.e global) statements about covariate predictors. Indeed, it is a necessary cautionary tale to observe that what may be robustly correlated with trust in one country (e.g. the religiosity effect) does not translate to the UK.

## Conclusions

While we can say little as to why self-assessed trust in science has gone up in the UK through the pandemic, we can conclude that trust in science is a variable that can be increased to substantial degrees and that it is highly likely that such increases in trust have been important for the effective public health response. Our data also indicate that the change in trust is more specifically attached to the relevant scientific groups. What is perhaps more disturbing is that positive changes in trust were largely restricted to those that were already trusting to some degree. There appears to have been a backfire effect [21] whereby those less trusting have become even less trusting, or at least so they report. This suggests that "one approach fits all" remedies for increasing trust [25] have the potential to do harm. That lack of trust appears to predict vaccine hesitancy renders this a question of public health as much as one of public understanding of science.

## Supporting information

**S1 File. The survey and its demographics.**
(PDF)

**S1 Fig. The relationship between political score and voting patterns in 2019 general election.**
(PDF)

## Acknowledgments

The authors wish to acknowledge discussion with Will Silcox and Momna Hejmadi in preparation of this manuscript. In addition, they wish to thank Anne Ferguson-Smith, Wendy Bickmore, Adam Rutherford, Patrick Sturgis and Sarah Cunningham-Burley for input into questionnaire design.

## Author Contributions

**Conceptualization:** Sofia Radrizzani, Cristina Fonseca, Alison Woollard, Jonathan Pettitt, Laurence D. Hurst.

**Data curation:** Sofia Radrizzani, Cristina Fonseca, Alison Woollard, Jonathan Pettitt, Laurence D. Hurst.

**Formal analysis:** Sofia Radrizzani, Laurence D. Hurst.

**Funding acquisition:** Cristina Fonseca, Alison Woollard.

**Investigation:** Sofia Radrizzani, Laurence D. Hurst.

**Methodology:** Sofia Radrizzani, Cristina Fonseca, Alison Woollard, Jonathan Pettitt, Laurence D. Hurst.

**Project administration:** Cristina Fonseca, Laurence D. Hurst.

**Resources:** Cristina Fonseca, Alison Woollard, Jonathan Pettitt.

**Software:** Sofia Radrizzani, Laurence D. Hurst.

**Supervision:** Laurence D. Hurst.

**Visualization:** Sofia Radrizzani, Laurence D. Hurst.

**Writing – original draft:** Sofia Radrizzani, Laurence D. Hurst.

**Writing – review & editing:** Sofia Radrizzani, Cristina Fonseca, Alison Woollard, Jonathan Pettitt, Laurence D. Hurst.

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
