## [Decision Letter · Decision Letter 0]

31 Oct 2022

PONE-D-22-25581Both trust in, and polarization of trust in, relevant sciences have increased through the COVID-19 pandemicPLOS ONE

Dear Dr. Laurence D. Hurst,

Thank you for submitting your manuscript to PLOS ONE. After careful consideration, we feel that it has merit but does not fully meet PLOS ONE’s publication criteria as it currently stands. Therefore, we invite you to submit a revised version of the manuscript that addresses the points raised during the review process.

We look forward to receiving your revised manuscript.

Kind regards,

Srikanth Umakanthan

Academic Editor

PLOS ONE

Journal Requirements:

3. Please include a copy of Table 6 which you refer to in your text on page 13.

4. Please review your reference list to ensure that it is complete and correct. If you have cited papers that have been retracted, please include the rationale for doing so in the manuscript text or remove these references and replace them with relevant current references. Any changes to the reference list should be mentioned in the rebuttal letter that accompanies your revised manuscript. If you need to cite a retracted article, indicate the article’s retracted status in the References list and also include a citation and full reference for the retraction notice.

Additional Editor Comments:

Minor revision recommended by the esteemed reviewers.

Reviewers' comments:

Reviewer's Responses to Questions

**Comments to the Author**

1. Is the manuscript technically sound, and do the data support the conclusions?

Reviewer #1: Yes

Reviewer #2: Yes

2. Has the statistical analysis been performed appropriately and rigorously? 

Reviewer #1: Yes

Reviewer #2: Yes

3. Have the authors made all data underlying the findings in their manuscript fully available?

Reviewer #1: Yes

Reviewer #2: Yes

4. Is the manuscript presented in an intelligible fashion and written in standard English?

Reviewer #1: Yes

Reviewer #2: Yes

5. Review Comments to the Author

Reviewer #1: Abstract: Needs to be curtailed and specific.

Introduction and discussion can be strengthened by including the following points:

1. include a sentence on origin, transmission of COVID-19 (refer and cite: doi: 10.1136/postgradmedj-2020-138234)

2. The role of government in combatting COVID-19 in comparison with the other regions (refer and cite: doi: 10.3389/fpubh.2022.844333).

3. The role of social environmental predictors of COVID-19 in general (refer and cite: doi: 10.3390/vaccines10101749)

4. The effect of vaccine hesitancy in COVID-19 and its effects in trust in science (refer and cite: doi: 10.3390/vaccines9101064)

5. The calibre of predictors in COVID-19 (refer and cite: doi: 10.1136/postgradmedj-2021-141365)

Materials and methods: Explain more in detail about the survey, the target population, inclusion and exclusion criteria, if any missed cases were included?

Statistics: divide the covariates into dependent and independent variables.

Mention if bias was generated, and if yes, how was it limited/regressed.

The importance of choosing the statistical relevance should be addressed.

Limitations and strengths need to be included in the end of discussion.

Reviewer #2: The manuscript should include more figures and illustrations.

To add more details about survey analysis.

Was the survey validated? was a pilot study done?if yes those details need to be included.

The regional comparison should be included.

6. PLOS authors have the option to publish the peer review history of their article (what does this mean?). If published, this will include your full peer review and any attached files.

Reviewer #1: No

Reviewer #2: No

---

## [Editor Report · Decision Letter 1]

11 Nov 2022

Both trust in, and polarization of trust in, relevant sciences have increased through the COVID-19 pandemic

PONE-D-22-25581R1

Dear Dr. Hurst,

We’re pleased to inform you that your manuscript has been judged scientifically suitable for publication and will be formally accepted for publication once it meets all outstanding technical requirements.

Kind regards,

Srikanth Umakanthan

Academic Editor

PLOS ONE

Additional Editor Comments (optional):

Accept in present revised format.
---

## [Editor Report · Acceptance letter]

15 Nov 2022

PONE-D-22-25581R1 

Both trust in, and polarization of trust in, relevant sciences have increased through the COVID-19 pandemic 

Dear Dr. Hurst:

I'm pleased to inform you that your manuscript has been deemed suitable for publication in PLOS ONE. Congratulations! Your manuscript is now with our production department. 

Kind regards, 

on behalf of

Dr. Srikanth Umakanthan 

Academic Editor

PLOS ONE